# Exploring Clinical Correlates of Metacognition in Bipolar Disorders Using Moderation Analyses: The Role of Antipsychotics

**DOI:** 10.3390/jcm10194349

**Published:** 2021-09-24

**Authors:** Paul Roux, Nathan Faivre, Anne-Sophie Cannavo, Eric Brunet-Gouet, Christine Passerieux

**Affiliations:** 1DisAP-DevPsy-CESP, UMR 1018, INSERM/Université Paris-Saclay/Université Versailles Saint-Quentin-En-Yvelines, 94807 Villejuif, France; ebrunet@ch-versailles.fr (E.B.-G.); cpasserieux@ch-versailles.fr (C.P.); 2Centre Hospitalier de Versailles, Service Hospitalo-Universitaire de Psychiatrie d’Adultes et d’Addictologie, 78150 Le Chesnay, France; ascannavo@ch-versailles.fr; 3Fondation Fondamental, 94000 Créteil, France; 4Univ. Grenoble Alpes, Univ. Savoie Mont Blanc, CNRS, LPNC, 38000 Grenoble, France; nathan.faivre@univ-grenoble-alpes.fr

**Keywords:** bipolar disorders, metacognition, cognitive complaints, cognition, antipsychotic, impulsivity

## Abstract

The determinants of metacognition are still poorly understood in bipolar disorders (BD). We aimed to examine the clinical determinants of metacognition, defined as the agreement between objective and subjective cognition in individuals with BD. The participants consisted of 281 patients with BD who underwent an extensive neuropsychological battery and clinical evaluation. To assess subjective cognition, participants provided a general rating of their estimated cognitive difficulties. Clinical characteristics of BD were also recorded, along with medication. We studied the potential moderation of the association between cognitive complaints and global objective cognitive performance by several clinical variables with ordinal logistic regressions. Depression and impulsivity were associated with greater cognitive complaints. The only variable that moderated the relationship between objective and subjective cognition in the global model was the prescription of antipsychotics. Patients taking antipsychotics had a poorer association between cognitive complaints and objective neuropsychological performance. This result suggests a role for dopamine in the modulation of metacognitive performance, and calls for the systematic control of antipsychotic medication in future studies documenting metacognitive deficits in severe and persistent mental disorders. Depression and impulsivity should be investigated as potential therapeutic targets for individuals with BD and cognitive complaints, before proposing an extensive neuropsychological evaluation.

## 1. Introduction

Metacognition refers to a spectrum of mental activities of which the object is one’s own thoughts. In individuals suffering from severe and persistent psychiatric disorders, metacognition is a multi-faceted construct encompassing the recognition that one is ill, the awareness of one’s own cognitive style and beliefs, and the awareness of one’s own cognitive performance [1]. Lack of insight into the illness is frequent in BD (Bipolar Disorders) [2], particularly during the manic phase [3]. Several studies have identified dysfunctional metacognitive beliefs in BD [4], which are associated with a higher level of depression, earlier onset of affective illness [5], and worse cognitive impairments [6]. Cognitive insight in BD is characterized by higher self-reflectiveness, i.e., the capacity to reflect on one’s own experiences, which correlates with more severe depression [7]. 

One way to evaluate the awareness of one’s cognitive performance is to measure the agreement between objective cognition measured using established neuropsychological tests and subjective cognition assessed using self-reported scales. In this framework, good metacognitive performance implies a close relationship between objective and subjective cognition, whereas metacognitive deficits are reflected by their decorrelation. Several studies have explored the association between objective and subjective cognition but reported inconsistent results. Some reported non-significant or weak (r < 0.3) associations [8,9], thus suggesting poor metacognitive performance in BD. In contrast, other studies reported a more pronounced association [10], which appeared to depend on how objective cognition was measured. A previous study indeed reported that the global self-concept of general cognitive ability did not correlate with the general cognitive composite score in BD, whereas self-reported performance just after completing a specific cognitive task correlated with the objective performance [11].

Several attempts have been made to identify the determinants of metacognition in BD. Some proposed computing a meta-sensitivity index as the difference between the individual ranks in objective and subjective cognition, and reported several factors that covaried with this index: mood symptoms, number of hospitalizations, type of BD, socio-occupational difficulties, perceived stress, quality of life, and verbal IQ [12]. Others subtracted the z-scores for objective cognition from the z-scores of subjective cognition. One study using this method found no clinical correlates of metacognition in BD [11], whereas another reported that metacognition was associated with objective cognitive performance [6]. However, the computation of a score reflecting a difference in ranks or z-scores between objective and subjective cognition makes it difficult to test whether a significant association between this score and a clinical correlate could be explained by a simpler and more direct relationship between this correlate and objective or subjective cognition.

An alternative strategy is to use moderation analysis, which avoids the limitation of using a difference between scores. One study reported that depressive symptoms did not moderate the association between subjective and objective cognition in euthymic or mildly depressed individuals with BD, thus suggesting a lack of association between depressive symptoms and metacognition in BD [9]. On the contrary, a later study reported that the association between subjective and objective cognition was also moderated by depressive symptoms only in individuals with depression and by manic symptoms only in individuals with hypomania/mania [13]. However, these two studies had a relatively limited sample size (*n* < 150), and investigated only mood symptoms as potential correlates of metacognition in BD, ignoring the potential role of important variables such as medication [14], psychosocial functioning [12] and childhood trauma [15].

Here, we aimed to identify potential clinical correlates of metacognition in a large sample of individuals with BD, beyond mood symptoms, using multiple moderation analysis between cognitive complaints and objective cognition.

## 2. Materials and Methods

### 2.1. Study Design and Characteristics of the Recruiting Network

This monocenter, transversal study included patients recruited into the FACE-BD (FondaMental Advanced Centers of Expertise for Bipolar Disorders) cohort from the BD Expert Center of Versailles. The study was pre-registered (NCT04034147). The authors assert that all procedures contributing to this work comply with the ethical standards of the relevant national and institutional committees on human experimentation and with the Helsinki Declaration of 1975, as revised in 2008. All procedures involving human patients were approved by the local ethics committee (Comité de Protection des Personnes Ile de France IX) on 18 January 2010 under French laws for non-interventional studies (observational studies without any risk, constraint, or supplementary or unusual procedure concerning diagnosis, treatment, or monitoring). The board required that all patients be given an informational letter but waived the requirement for written informed consent. However, verbal consent was witnessed and formally recorded. Anonymized data are stored in a national database that was approved by the French body overseeing the safety of computerized databases (Commission Nationale de l’Informatique et des Liberte, DR-2011-069). No incentive was given to patients for their participation.

### 2.2. Participants

General practitioners or psychiatrists referred the patients who were assessed in the Centers of Expertise. The goal of this referral was to improve global care by delivering personalized care plans, after a systematic set of comprehensive assessment tools including a neuropsychological battery. Patients were interviewed by senior psychiatrists, psychologists, or neuropsychologists specialized in bipolar disorders, who were all members of the specialized multidisciplinary teams of the Expert Centres. Senior psychiatrists diagnosed bipolar disorders with the SCID-IV (Structured clinical interview for DSM-IV-TR) [16]. Outpatients with type 1, type 2, or NOS (Not otherwise specified) BD, who were between 18 and 65 years of age, were eligible for this analysis. Patients who met the following criteria were excluded: history of neurological disorder, dyslexia, dysorthographia, dyscalculia, dysphasia, dyspraxia, substance-related disorders in the previous month (except tobacco use), or electroconvulsive therapy in the past year. No criteria related to the current mood state at inclusion were used to preserve the variability of levels of objective and subjective cognition. However, individuals whose symptom intensity was judged to be incompatible with the one-and-a-half-day evaluation were excluded (for instance, high suicidal risk, agitation, severe distractibility, disability to think or concentrate, or severe indecisiveness). Neuropsychologists screened exclusion criteria related to neurodevelopmental disorders; senior psychiatrists screened exclusion criteria related to neurological and substance-use disorders.

### 2.3. Assessment Tools

The sociodemographic variables collected at inclusion were sex, age, and education level.

#### 2.3.1. Clinical Assessments

The following clinical variables were recorded using the SCID: age at onset of BD, number and type of previous mood episodes, the subtypes of BD, and history of psychotic symptoms. Predominant polarity was determined following previous recommendations [17].

The CGI-S (Clinical Global Impression-Severity) scale assessed illness severity [18]. We used a yes/no questionnaire for recording patient treatment at the time of evaluation: lithium carbonate, anticonvulsants, antipsychotics, antidepressants, or anxiolytics. Mania was measured using the YMRS (Young Mania Rating Scale) total score [19] and depression using the MADRS (Montgomery Asberg Depression Rating Scale) total score [20]. The state of anxiety was measured using the total score of the state subscale of the STAI-Y-A (State-Trait Anxiety Inventory, form Y-A) [21]. Impulsivity was assessed using the total score of the BIS-10 (Barratt Impulsiveness Scale version 10) [22]. Childhood traumatic events were recorded using the total score of the CTQ (Childhood Trauma Questionnaire) [23]. Domain-based psychosocial functioning was measured using the FAST (Functioning Assessment Short Test) [24]. In this study, the total FAST score was used (higher score meaning poorer functioning). Adherence to medication was measured using the total score of the MARS (Medication Adherence Rating Scale) [25]. According to previous reports, all these tools have good psychometric properties (see Appendix A) for a description of the internal consistency and test-retest reliability for each tool).

#### 2.3.2. Objective and Subjective Cognition

Objective cognition: the battery of cognitive tests

The standardized test battery included 11 tests, amongst which five were subtests from the WAIS (Wechsler Adult Intelligence Scale) version III [26] or version IV [27]. Participants were included between June 2009 and December 2018. The French version of the WAIS-IV became available during this period, and we decided to start to use this version in May 2012. The rationale behind this decision was to use the most recent normative data available, as they gave a more accurate picture of the distance in cognitive performances between the recruited participants and the norm. The battery evaluated six domains:-**Processing speed**: Digit symbol coding (WAIS-III) or coding (WAIS-IV), WAIS symbol search, and TMT (Trail-Making Test) [28] part A-**Verbal memory**: California Verbal Learning Test [29] short and long delay free recall and total recognition-**Attention**: Conners’ Continuous Performance Test II [30] (detectability)-**Working memory**: WAIS digit span (total score) and spatial span (forward and backward scores) from the Wechsler Memory Scale version III [31]-**Executive functions**: color/word condition of the Stroop test [32], semantic and phonemic verbal fluency [33] and TMT part B-**Verbal and perceptual reasoning**: WAIS vocabulary and matrices

Raw scores were transformed to demographically corrected standardized z-scores based on normative data [30,32,34,35]. Higher scores reflected better performance. We computed a mean score for each cognitive domain. Then we computed a global score for objective cognition by averaging the cognitive domain scores. WAIS-III and WAIS-IV standardized scores were analysed together, as this method has been used in several cohort studies [36,37,38,39,40].

Subjective cognition: cognitive complaints

Cognitive complaints were assessed with item 10 of the QIDS-SR16 (Quick Inventory of Depressive Symptomatology—Self-Report): “During the past seven days, there has been no change in my usual capacity to concentrate or make decisions” (scored 0); “I occasionally feel indecisive or find that my attention wanders” (scored 1); “Most of the time, I struggle to focus my attention or to make decisions” (scored 2); “I cannot concentrate well enough to read or cannot make even minor decisions” (scored 3). Previous studies have reported evidence of validity and reliability of this single-item assessment. Its test-retest reliability over six months is good (Cronbach alpha of 0.74) [41]. Concurrent validity for this item has been investigated against a similar item rated by a clinician within the clinical-rated form of QIDS, and it was found to be satisfactory [42]. Finally, we checked the concurrent validity of this item within our sample. We ran a correlation analysis between item 10 of QIDS and the cognitive functioning factor of FAST, which assesses the subjective level of cognitive functioning perceived by the clinician. We found a moderate-to-strong correlation (Pearson correlation coefficient of 0.48, t = 9.148, df = 273, *p*-value < 0.001), thus suggesting a good concurrent validity of the single-item assessment of cognitive complaints used in this study.

### 2.4. Statistical Analyses

First, missing data were estimated using multivariate imputations by chained equations (50 imputations, mice package of R). The fraction of missing information (fmi) and the proportion of total variance due to missingness (λ) are reported in the results.

Metacognition was quantified as the strength of the association between cognitive complaints and the average cognitive performance on the neuropsychological battery: a negative and significant association between cognitive complaints and objective cognition was interpreted as good metacognition, whereas a lack of an association or a positive association was interpreted as impaired metacognition. This association was operationalized through an ordinal logistic regression with cognitive complaints as the dependent variable and objective cognition as the independent variable. We first ran successive moderation analyses with cognitive complaints as the dependent variable and two independent variables (one clinical moderator belonging to the variables listed above and objective cognition) by declaring their main effects and their interaction in the model. Assuming that metacognition reflects the strength of the association between objective cognition and subjective complaints, a variable was interpreted as a potential moderator of metacognition if its interaction with objective cognition was significant.

We then ran a multiple ordinal logistic regression with several independent variables to check whether the potential clinical moderator identified in the simple logistic regressions remained significant while simultaneously accounting for all the effects. Beyond the main effect of objective cognition, the independent variables were included in this multiple model if:-the level of significance of their interaction with objective cognition was *p* < 0.25. This threshold is usual for selecting variables for multiple regression [43,44]. In this case, both the interaction term and the main effect of the clinical covariate was included in the model-the level of significance of their interaction with objective cognition was *p* ≥ 0.25 and the level of significance of their main effect was *p* < 0.25. In this case, only the main effect of the clinical covariate was included in the model

## 3. Results

### 3.1. Characteristics of Participants

We included 281 participants. Socio-demographic, clinical, and functional characteristics of the sample are presented in Table 1.

The participants consisted mostly of women with type 2 BD and indeterminate polarity of mood episodes. Most participants were not in an active characterized mood episode during the evaluation: only 13.9% were currently undergoing a major depressive episode. None were manic and only 1.4% were hypomanic. The average severity of illness was between moderate and marked (4.5 ± 0.9).

The MADRS correlated above |r| > 0.5 with the end of the last characterized episode <3 months, the presence of a current major depressive episode, and the state subscale of STAI-YA. These last three variables were thus discarded from the following analyses to avoid multicollinearity issues in the multiple regression. Due to the small sample size, the NOS diagnosis (*n* = 30) was combined with that of type 2 BD (its nearest neighbor, *n* = 146) and manic polarity (*n* = 22) with the indeterminate class (its nearest neighbor, *n* = 86) of predominant mood polarity.

The neuropsychological results are presented in Table 2.

WAIS-III was proposed to 36 participants, and WAIS-IV to 245 participants. The worst performance in objective cognition was found for attention (−0.4 ± 0.7) and the best for reasoning (0.6 ± 0.7).

For cognitive complaints, the distribution of answers to item 10 of the QIDS was the following (see Appendix A): 31.8% for “No change in usual capacity to concentrate and decide“; 34.3% for “Occasionally feels indecisive or notes that attention often wanders”; 20.9% for “Most of the time struggles to focus attention or make decisions”; and 13.0% for “Cannot concentrate well enough to read or cannot make even minor decisions”. The mean cognitive complaints score was 1.2 (sd:1, *n* = 277), suggesting occasional subjective cognitive difficulties on average in this sample.

Imputation diagnostic is reported in Appendix A. Data were considered missing at random, as no systematic relationship between domains of missingness and participants’ socio-demographic characteristic were found.

### 3.2. Moderation Analyses

The bivariable ordinal logistic regression of cognitive complaints on objective cognition was not significant (OR = 0.70 (0.44–1.1), t(257.5) = −1.6, *p* = 0.113, λ = 0.051, fmi = 0.058). Overall, the study sample showed a non-significant association between objective and subjective cognition.

#### 3.2.1. Trivariable Ordinal Logistic Regressions

The results of the ordinal logistic regressions of cognitive complaints with objective cognition and several successive clinical moderators as independent variables are presented in Table 3 for the interaction between objective cognition and clinical moderators, and Appendix A for the main effect of clinical moderators and objective cognition respectively.

Significant interactions with objective cognitive performance were found for predominant mood polarity and antipsychotic medication (see Table 3). The distribution of objective cognition according to the level of cognitive complaints and polarity in the observed dataset are reported in Appendix A. The association between objective cognition and cognitive complaints was significant for depressive polarity (OR = 0.33 (0.13–0.80), t(66.6) = −2.5, *p* = 0.015, λ = 0.224, fmi = 0.246), but not indeterminate/manic polarity (OR = 1.02 (0.56–1.87), t(132.6) = 0.1, *p* = 0.947, λ = 0.179, fmi = 0.192). The distribution of objective cognition according to the level of cognitive complaints and the prescription of antipsychotics in the observed dataset are presented in Figure 1.

The association between objective cognition and cognitive complaints was significant for patients not taking antipsychotics (OR = 0.48 (0.28–0.78), t(215.8) = −2.9, *p* = 0.004, λ = 0.048, fmi = 0.057), but not for those taking antipsychotics (OR = 2.82 (0.73–10.8), t(36.5) = 1.6, *p* = 0.127, λ = 0.086, fmi = 0.132).

The clinical moderators that were significantly associated with more cognitive complaints in the absence of a significant interaction with objective cognition were a type 2/NOS BD (OR = 1.75 (1.11–2.76), t(269.6) = 2.4, *p* = 0.017), a lack of history of psychosis (OR = 1.80 (1.01–3.22), t(177.1) = −2, *p* = 0.047), greater illness severity measured based on the CGI-S (OR = 1.84 (1.38–2.44), t(267.9) = 4.2, *p* < 0.001), more severe depressive symptoms measured based on the MADRS (OR = 1.14 (1.1–1.17), t(265.6) = 8.4, *p* < 0.001), anxiolytic (OR = 2.12 (1.22–3.67), t(270.6) = 2.7, *p* = 0.008), a lifetime substance use disorder (OR = 1.7 (1.02–2.83), t(259.4) = 2, *p* = 0.043), greater impulsivity measured based on the BIS (OR = 1.08 (1.06–1.1), t(268.4) = 6.7, *p* < 0.001), an history of trauma assessed based on the CTQ (OR = 1.04 (1.02–1.05), t(261.2) = 4.7, *p* < 0.001), poorer functioning measured based on the FAST (OR = 1.08 (1.06–1.1), t(269.5) = 8.1, *p* < 0.001) and poorer medication adherence measured based on the MARS (OR = 0.87 (0.78–0.96), t(234.2) = −2.7, *p* = 0.008, see Appendix A).

#### 3.2.2. Multiple Ordinal Logistic Regression

The multiple ordinal logistic regression included cognitive complaints as the dependent variable and the following independent variables: predominant mood polarity, CGI-S, lithium carbonate, antipsychotic, anxiolytic and FAST (their main effect and their interaction with objective cognition), and the main effect of objective cognition, educational level, type of BD, history of psychosis, MADRS, any lifetime substance use disorder, BIS, CTQ, MARS and type of WAIS (see Table 4).

The only interaction with objective cognition that remained significant in the multiple moderation analysis was found for antipsychotics (OR = 6 (1.13–31.7), t(216.9) = 2.1, *p* = 0.035). The variables which were significantly associated with increased cognitive complaints were MADRS (OR = 1.1 (1.06–1.14), t(240.5) = 5.2, *p* < 0.001) and BIS (OR = 1.07 (1.04–1.1), t(243.5) = 4.4, *p* < 0.001).

## 4. Discussion

We aimed to identify clinical correlates of metacognition in BD using moderation analysis between cognitive complaints and objective cognition.

First, this study confirmed the weak correlation between objective performance on a battery of neuropsychological tests and subjective perception of cognitive functioning reported in previous studies [8]. Among all investigated clinical variables, only the predominant mood polarity and the prescription of antipsychotics influenced the strength of the association between cognitive complaints and objective cognition. In the absence of covariates, the association between objective cognition and cognitive complaints was more robust for the depressive polarity than for the indeterminate/manic polarity. A negative impact of manic episodes on objective cognition has been reported previously [45], which may progressively lead to an impairment of metacognition. An alternative interpretation is that preserved metacognition facilitates the emergence of depression, whereas impaired metacognition may lead to disinhibition and mania. However, the moderating effect of polarity did not resist the introduction of covariates in the model.

The main factor influencing the strength of the association between cognitive complaints and objective cognition was the prescription of antipsychotics, for which the moderating effect remained significant in the multiple analysis. This result is compatible with the hypothesis that confidence in cognitive performance is modulated by dopamine [46]. More specifically, our results suggest that dopamine antagonists, such as antipsychotics, may decrease metacognitive accuracy. A previous study reported that dopamine administration increased metacognition in healthy participants, in parallel with increased amplitudes of MEG oscillations in the medial prefrontal cortex [47]. Another study recently demonstrated that the administration of haloperidol, a dopamine antagonist, impaired metamemory in healthy individuals in parallel with aberrant fMRI activity in frontostriatal circuits [14]. Of note, this effect may depend on the antipsychotic investigated, as it was not replicated with amisulpride [48]. This is the first study to report an association between antipsychotic medication and metacognition in BD. However, it is not possible to conclude a causal link of antipsychotics on metacognitive impairments, as we did not investigate the dose effect of antipsychotics on metacognition in this study. Longitudinal studies are needed to clarify the effect of antipsychotics on metacognition in BD and should account for the specific psychopharmacological properties of a particular antipsychotic, along with the daily dosage and serum level, duration of exposure, and therapeutic response. Despite these limitations, our result strongly encourages controlling for antipsychotic medication in studies investigating metacognition in severe and persistent psychiatric disorders.

Our results do not support a moderating effect of mood symptoms on the relationship between objective and subjective cognition in BD. This result is consistent with those of previous reports showing a lack of association between mood symptoms and metacognition in euthymic [9,11] and mildly depressed [6] individuals with BD. In contrast, mood symptoms were associated with a weaker relationship between cognitive complaints and objective cognitive performance in acutely ill individuals with BD [13]. This apparent discrepancy between studies can be explained by heterogeneity in the level of mood symptoms, which were low in our sample. Our results did not replicate the findings that hospitalizations and more significant socio-occupational difficulties correlate with metacognition, which were associated with disproportionately more subjective complaints than objective impairment in a previous report involving remitted individuals with BD [12].

The participants’ profile might have influenced some of our results, which may not generalize to all individuals with BD. First, there was a majority of BD 2 (52%) compared with BD 1 (37%). We did not find in the present study a significant association between the type of BD and metacognition. However, a previous study has reported that BD type II was associated with having more subjective complaints than objective impairment [12]. Thus, prevalence of BD type II in this sample might have biased recruiting toward individuals who underestimated their cognitive performance. Second, patients had very low manic symptoms. The majority of participants were euthymic or slightly depressed. Again, this might have biased the recruitment toward excluding individuals who overestimated their cognitive performance and lacked insight.

The multiple regression analysis also identified two robust correlates of cognitive complaints: depressive symptoms and impulsivity. Greater depressive symptoms have been consistently identified as important determinants of cognitive complaints in BD [8,9,13] along with less strong manic symptoms [13]. The association between impulsivity and cognitive complaints may be explained by symptoms of attention deficit hyperactivity disorder [49], which were not investigated in our sample. Our results are consistent with those of a previous study reporting a lack of greater cognitive complaints by individuals with BD taking lithium or antipsychotic medication [10]. Our results did not replicate the findings that a higher number of episodes, especially the number of mixed episodes, longer duration of the illness and onset of the illness at an earlier age [50], and impairment in psychosocial functioning [51] are associated with more subjective complaints in BD.

The most important limitation was related to the assessment of subjective cognition, which was based only on one question extracted from a scale not specifically designated to measure cognitive complaints. Further studies should be conducted to replicate our results using a scale validated in BD, such as the Cognitive Complaints in Bipolar Disorder Rating Assessment [52]. The measure of objective cognition used in this study was able to identify performance above the norm, whereas the assessment of subjective cognition could not measure better self-reported performance, as cognitive complaints have a low boundary corresponding to normal performance. Moreover, the cognitive complaints question used in this study evaluated subjective cognitive performance relative to the usual capacity of the patient, whereas the measure of objective cognition referred to the distance to a norm of healthy subjects. Others have proposed using a different question to assess subjective cognition in order to resolve these two issues: “compared to healthy individuals of your age, your cognitive skills (concentration, memory, problem-solving, …) are profoundly below average, well below average, below average, average, above average, well above average, or superior” [6,11]. As a consequence, the way we measured subjective cognition in this study may have led to an underestimation of the strength of the association with objective cognition. Another important limitation was the investigation of global cognition only, without exploring the correlates of metacognition within each specific cognitive domain. This may be problematic, as metacognition varies by the domain of impairment, with patients being particularly unaware of attention and processing speed problems [12]. Further studies investigating metacognition in BD may benefit from less biased measures such as the ratio between meta-d’ and d [53]. The participants included in the present study were heterogeneous regarding their mood symptoms, mixing participants with a characterized depressive episode and euthymic participants. Further studies should control mood symptoms by recruiting separated subgroups with euthymic, hypomanic and depressed participants with equal sample size. Finally, grouping antipsychotics in just one class was an important limitation of the current study. Some antipsychotics are known to have a potential direct procognitive effect, such as lurasidone [54], and an indirect procognitive effect through an antidepressant action (ex: quetiapine, lurasidone).

## 5. Conclusions

This exploratory study investigated several clinical correlates of metacognition in BD using moderation analysis. The main factor influencing metacognition was the prescription of antipsychotics. The strength of the relationship between cognitive complaints and objective cognitive performance was weaker for individuals taking an antipsychotic. This result suggests a crucial role of dopamine in the modulation of metacognitive performance. Our study emphasizes the importance of controlling for antipsychotic medication when assessing metacognition in severe and persistent mental disorders, such as schizophrenia and BD, in future studies. This result also suggests not to rely on the sole self-report evaluation of cognitive functioning in patients with BD who are taking antipsychotics, and complete this evaluation with objective measures of cognitive performance. Depressive symptoms and impulsivity were associated with poorer subjective cognition and may be considered as potential therapeutic targets for individuals with BD and cognitive complaints. Our results may also guide future programs of metacognitive training in BD.

## Figures and Tables

**Figure 1 jcm-10-04349-f001:**
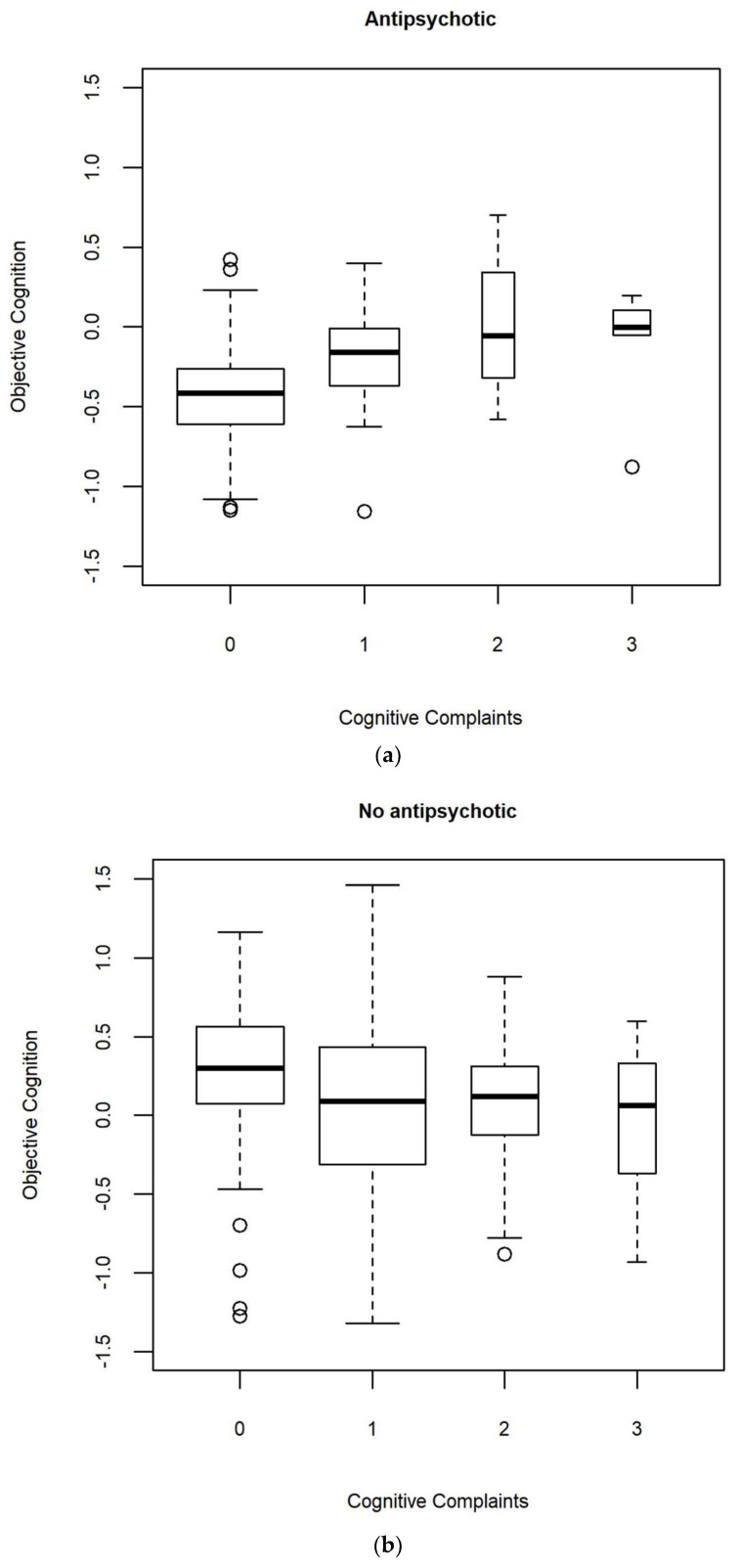
Distribution of objective cognition according to the level of cognitive complaints in the observed dataset: (**a**) in individuals with antipsychotic; (**b**) in individuals without antipsychotic. The width of the boxes is proportional to the sample size in each level of cognitive complaints.

**Table 1 jcm-10-04349-t001:** Participants’ socio-demographical, clinical, and functional characteristics.

Variable	Mean or %	SD	*n*
Age (years)	40.2	11.1	281
Sex	42.3 (M)		281
Educational level (years)	14.3	2.5	281
Diagnosis: Type 1	37.4		281
Type 2	52		
NOS ^1^	10.6		
Total number of mood episodes	8.1	7.5	170
Predominant polarity: depressive	36.5		170
Indeterminate	50.6		
Manic	12.9		
Age at onset (years)	23.3	8.2	271
History of psychosis	19.8		232
Rapid cycling	7.6		263
CGI ^2^ Severity (1–7)	4.5	0.9	279
Current major depressive episode	13.9		281
Current hypomanic episode	1.4		281
Current manic episode	0		281
MADRS ^3^ (0–60)	10.4	8.5	281
YMRS ^4^ (0–60)	2.2	3.4	281
STAI-YA ^5^ (state subscale) (20–80)	43.4	14.5	278
End of last characterized episode > 3 months	55.6		279
Antidepressant	22.1		281
Anticonvulsant	25.3		281
Lithium Carbonate	13.5		281
Antipsychotic	16.4		281
Anxiolytic	19.2		281
Any lifetime substance use disorder	26		273
BIS ^6^ (34–136)	66.9	10.6	279
CTQ ^7^ (28–140)	43.7	15.1	277
FAST ^8^ total (0–72)	20	13	280
MARS ^9^ (0–10)	6.8	2.2	254

^1^ NOS: not otherwise specified; ^2^ CGI: Clinical Global Impression scale; ^3^ MADRS: Montgomery Asberg Depression Rating Scale; ^4^ YMRS: Young Mania Rating Scale; ^5^ STAI-YA: State-Trait Anxiety Inventory YA form; ^6^ BIS: Barratt Impulsiveness Scale; ^7^ CTQ: Childhood Trauma Questionnaire; ^8^ FAST: Functioning Assessment Short Test; ^9^ MARS: Medication Adherence Rating Scale.

**Table 2 jcm-10-04349-t002:** Objective neuropsychological performances expressed in standard deviations from the norm.

	Mean	SD	*n*
**Verbal Memory**	**0.41**	**0.9**	
CVLT ^1^ Immediate recall	0.58	1.26	279
CVLT ^1^ Short delay free recall	0.29	1.11	279
CVLT ^1^ Long delay free recall	0.37	1.09	278
CVLT ^1^ Total recognition	0.38	0.59	278
**Working Memory**	**−0.14**	**0.7**	
Digit Span	−0.24	0.86	273
Spatial Span forward	−0.08	0.88	270
Spatial Span backward	−0.12	0.85	270
**Executive Functioning**	**−0.15**	**0.77**	
TMT ^2^ Part B	−0.14	1.18	279
Stroop color/word condition	0.04	1.02	267
Phonemic fluency	−0.07	1.05	278
Semantic fluency	−0.42	0.96	278
**Processing speed**	**−0.11**	**0.67**	
Coding	−0.2	0.92	271
Symbol search	0.01	0.88	270
Stroop word condition	−0.08	0.79	268
Stroop color condition	−0.45	0.85	267
TMT ^2^ Part B	0.18	0.96	280
**Attention**	**−0.38**	**0.67**	
CPT ^3^ omission	−0.84	1.23	266
CPT ^3^ commission	−0.15	1.06	266
CPT ^3^ variability	−0.29	1.1	266
CPT ^3^ detectability	−0.23	0.98	266
**Reasoning**	**0.56**	**0.71**	
Vocabulary	0.83	0.87	251
Matrices	0.34	0.79	270

^1^ CVLT: California Verbal Learning Test; ^2^ TMT: Trail Making Test; ^3^ CPT: Continuous Performance Test.

**Table 3 jcm-10-04349-t003:** Results for the trivariable ordinal logistic regressions with cognitive complaints as the dependent variable and objective cognition, several successive clinical moderators, and the interaction between objective cognition and the clinical moderators as independent variables. This table reports only the interaction effect between objective cognition and the clinical moderator (the main effect of objective cognition and clinical moderator are reported in Appendix A).

Interaction between the Mean Cognitive Performance and a Clinical Moderator	OR (95% CI) ^1^	Statistic	*p*	λ	fmi
Age	0.98 (0.94–1.02)	t(250.5) = −0.9	0.392	0.063	0.071
Sex	1 (0.41–2.45)	t(258.3) = 0	0.992	0.044	0.051
Educational level	1.03 (0.86–1.24)	t(242.9) = 0.3	0.734	0.081	0.088
Diagnosis (Type 2/NOS ^2^ vs. Type 1)	0.83 (0.31–2.19)	t(245) = −0.4	0.706	0.076	0.083
Total number of mood episodes	0.98 (0.89–1.07)	t(72.7) = −0.5	0.599	0.536	0.548
Predominant Polarity (Indeterminate/Manic vs. Depressive)	3.24 (1.02–10.34)	t(128.4) = 2	0.047	0.335	0.345
Age at onset	1 (0.94–1.06)	t(251.1) = −0.1	0.896	0.062	0.069
History of psychosis	1.14 (0.32–4.02)	t(159.2) = 0.2	0.838	0.257	0.266
Rapid cycling	1.33 (0.25–6.93)	t(211.8) = 0.3	0.738	0.146	0.154
CGI ^3^ Severity	1.61 (0.88–2.94)	t(241.6) = 1.5	0.124	0.084	0.091
MADRS ^4^	1.01 (0.95–1.07)	t(252.2) = 0.2	0.826	0.059	0.067
YMRS ^5^	1.02 (0.89–1.17)	t(239.9) = 0.3	0.758	0.087	0.095
Antidepressant	0.82 (0.28–2.45)	t(251.5) = −0.4	0.724	0.061	0.068
Anticonvulsant	1.05 (0.31–3.63)	t(229.8) = 0.1	0.933	0.109	0.116
Lithium Carbonate	0.45 (0.12–1.69)	t(257.5) = −1.2	0.233	0.046	0.054
Antipsychotic	6.87 (1.64–28.76)	t(240) = 2.7	0.009	0.087	0.094
Anxiolytic	0.49 (0.17–1.47)	t(247.2) = −1.3	0.205	0.071	0.078
Any lifetime substance use disorder	1.15 (0.4–3.33)	t(234.1) = 0.3	0.793	0.1	0.107
BIS ^6^	1.02 (0.98–1.07)	t(249.4) = 1	0.309	0.066	0.073
CTQ ^7^	0.99 (0.96–1.03)	t(229.7) = −0.5	0.617	0.109	0.117
FAST ^8^	1.04 (1–1.08)	t(246.6) = 1.9	0.064	0.072	0.08
MARS ^9^	1.09 (0.89–1.33)	t(242.8) = 0.8	0.397	0.081	0.088
Type of WAIS ^10^	0.92 (0.23–3.7)	t(192.2) = −0.1	0.908	0.186	0.194

^1^ Odds Ratio [95% Confidence Interval]; ^2^ NOS: not otherwise specified; ^3^ CGI: Clinical Global Impression scale; ^4^ MADRS: Montgomery Asberg Depression Rating Scale; ^5^ YMRS: Young Mania Rating Scale; ^6^ BIS: Barratt Impulsiveness Scale; ^7^ CTQ: Childhood Trauma Questionnaire; ^8^ FAST: Functioning Assessment Short Test; ^9^ MARS: Medication Adherence Rating Scale; ^10^ Wechsler Adult Intelligence Scale.

**Table 4 jcm-10-04349-t004:** Results for the multiple moderation analysis with ordinal logistic regression, including cognitive complaints as the dependent variable.

Independant Variable	OR (95% CI) ^1^	Statistic	*p*	λ	fmi
Predominant Polarity (Indeterminate/Manic vs. Depressive)	1.17 (0.6–2.29)	t(116) = 0.5	0.646	0.353	0.364
CGI Severity	1.37 (0.96–1.96)	t(236.4) = 1.7	0.083	0.055	0.063
Lithium carbonate	1.42 (0.66–3.06)	t(248.9) = 0.9	0.366	0.019	0.026
Antipsychotic	1.4 (0.63–3.1)	t(240.5) = 0.8	0.408	0.044	0.052
Anxiolytic	1.89 (1–3.57)	t(248.7) = 2	0.051	0.019	0.027
FAST	1.02 (1–1.05)	t(245.8) = 1.9	0.058	0.028	0.036
Objective cognition	0.05 (0–1.58)	t(213.8) = −1.7	0.09	0.111	0.119
Educational level	0.97 (0.88–1.08)	t(247.5) = −0.5	0.596	0.023	0.031
Diagnosis (Type 2/NOS ^2^ vs. Type 1)	1.41 (0.67–2.98)	t(187.3) = 0.9	0.363	0.17	0.179
History of psychosis	1.22 (0.49–3.02)	t(121.2) = 0.4	0.664	0.337	0.348
**MADRS** ^3^	**1.1 (1.06–1.14)**	**t(240.5) = 5.2**	**<0.001**	**0.044**	**0.052**
Any lifetime substance use disorder	0.81 (0.45–1.45)	t(231.2) = −0.7	0.48	0.069	0.077
**BIS** ^4^	**1.07 (1.04–1.1)**	**t(243.5) = 4.4**	**<0.001**	**0.035**	**0.043**
CTQ ^5^	1.02 (1–1.03)	t(245.2) = 1.8	0.067	0.03	0.038
MARS ^6^	0.93 (0.82–1.06)	t(196.3) = −1.1	0.264	0.15	0.159
Type of WAIS ^7^ (IV vs. III)	0.75 (0.36–1.59)	t(246.1) = −0.7	0.458	0.027	0.035
Objective cognition: Polarity (Indeterminate/Manic vs. Depressive)	1.42 (0.37–5.47)	t(126.5) = 0.5	0.605	0.321	0.332
Objective cognition: CGI ^8^	1.52 (0.73–3.17)	t(228.2) = 1.1	0.265	0.076	0.084
Objective cognition: Lithium carbonate	0.49 (0.09–2.67)	t(221.4) = −0.8	0.405	0.093	0.101
**Objective cognition: Antipsychotic**	**6 (1.13–31.7)**	**t(216.9) = 2.1**	**0.035**	**0.104**	**0.112**
Objective cognition: Anxiolytic	0.52 (0.13–2.13)	t(209.5) = −0.9	0.36	0.121	0.129
Objective cognition: FAST ^9^	1.03 (0.98–1.08)	t(224.6) = 1.3	0.199	0.085	0.093

^1^ Odds Ratio [95% Confidence Interval]; ^2^ NOS: not otherwise specified; ^3^ MADRS: Montgomery Asberg Depression Rating Scale; ^4^ BIS: Barratt Impulsiveness Scale; ^5^ CTQ: Childhood Trauma Questionnaire; ^6^ MARS: Medication Adherence Rating Scale; ^7^ Wechsler Adult Intelligence Scale; ^8^ CGI: Clinical Global Impression scale; ^9^ FAST: Functioning Assessment Short Test. Significant results (*p* < 0.005) are in bold.

## Data Availability

The data presented in this study are available on request from the corresponding author. The data are not publicly available as participants did not give their consent for sharing these data.

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
