# Peer review of "Exploring Clinical Correlates of Metacognition in Bipolar Disorders Using Moderation Analyses: The Role of Antipsychotics"

_jcm, 2021, doi:10.3390/jcm10194349_

Round 1
Reviewer 1 Report
This observational study of metacognition among individuals with bipolar disorder is potentially an important contribution to the field, and can be extremely helpful to prescribers in determining the need for or selecting antipsychotics for these patients. There are, however, a number of methods/analytic concerns that should be addressed so that the conclusions can be clearer.
- For each instrument used, describe the validity/reliability of the instrument. This is especially important for the single item assessment of cognitive complaints; has this item been so used in previous studies? Is there evidence of validity and reliability for this single item? In the discussion the authors note that metacognition may vary by domain of functioning, but this item does not appear to account for that. This aspect of the methods needs to be defended with a strong rationale.
- Clarify how participants were recruited, how inclusion/exclusion criteria were assessed, and if participants where provided incentives for participation.
- It was an excellent decision for the authors to carefully describe imputation of missing data. Please clarify how much and what types of data were missing, and if there was any systematic relationship between domains of missing data and participant demographics.
- Clarify why the WAIS-III was used with some participants, and the WAIS-IV was used with others, and provide a rationale for using these differing instruments in the same analysis.
- I would prefer to see beta, rather than B's reported, as this would provide much clearer information on comparative strengths of predictors.
- Given the large numbers of both variables and analyses, please explicitly discuss both efforts to control for Type I error and observed power
- Minor note: on page 10, I believe that "antypsychotic" should read "antipsychotic."
Reviewer 2 Report
The study “Exploring clinical correlates of metacognition in bipolar disor- 2 ders using moderation analyses: the role of antipsychotics” adresses an interesting topic and is very well performed. Only a few minor issues should be adressed prior to publication.
The main limitation of the stydy, which is the measure of subjective cognition is already adequately discussed however some other limitations should be further elaborated. My main concern is the sample characteristics. Due to a small proportion of patients with type 1 BD or manic/hypomanic simptomatology, the results of this study basically represent the relationship between metacognition and the studied variables in type 2 BD or patients with a predominance of depression. This limitation should be further explained since it is very plausible that results would be very different with a higher proportion of patients with type 1 BD and manic predominance. Therefore extrapolation of the results to all BD patients should not be attempted.
Round 2
Reviewer 1 Report
The authors have addressed all of my concerns and I feel that this paper is now ready for publication.